# A Rare Case of Disseminated *Nocardia transvalensis* in an Immunocompetent Host

**DOI:** 10.3390/idr17060140

**Published:** 2025-11-12

**Authors:** Branavan Ragunanthan, Kevin Wunderly, James Kleshinski, Caitlyn Hollingshead

**Affiliations:** 1Department of Medicine, Infectious Disease, University of Toledo College of Medicine and Life Sciences, Toledo, OH 43614, USA; caitlyn.hollingshead@utoledo.edu; 2Department of Medicine, University of Toledo College of Medicine and Life Sciences, Toledo, OH 43614, USA; kevin.wunderly@utoledo.edu (K.W.); james.kleshinski@utoledo.edu (J.K.); 3Department of Medical Education, University of Toledo College of Medicine and Life Sciences, Toledo, OH 43614, USA

**Keywords:** *Nocardia transvalensis*, disseminated nocardia, paraspinal abscess

## Abstract

**Background**: *Nocardia* are a group of bacteria known to cause pulmonary, cutaneous, neurologic, or disseminated diseases, usually in immunocompromised hosts. Within the *Nocardia* family is *Nocardia transvalensis*, a rarely encountered and underreported organism in the clinical literature. **Case**: Here, we report the case of an immunocompetent patient presenting with lumbar pain diagnosed and treated for disseminated *Nocardia transvalensis* infection. Our patient underwent magnetic resonance imaging (MRI), demonstrating possible abscess and subtle osteomyelitis of the L3-L4 facet joint and transverse process; a subsequent biopsy and culture resulted in *Nocardia transvalensis*. Further imaging with a computed tomography (CT) scan of the head revealed a 9 mm enhancing supratentorial lesion. The patient was treated with empiric antibiotics, but this was narrowed to levofloxacin, linezolid, and trimethoprim-sulfamethoxazole after antibiotic sensitivities cropped up. **Conclusions**: Within this case, we extensively discuss the clinical pathogenesis of *Nocardia transvalensis* in an unusual host, the diagnostic approach to confirming active *Nocardia* infection, and the susceptibility patterns in a relatively unstudied organism.

## 1. Introduction

*Nocardia* are a group of bacteria that are rarely encountered clinically through the manifestation of a broad spectrum of diseases, often in a select, unique patient population. *Nocardia transvalensis* is a species of *Nocardia* that has seldom been observed in clinical practice, and its pathogenesis and disease management are rarely described in the medical literature [1,2,3]. In this section, we describe the microbiological features, epidemiology, known transmission, diagnosis, and spectrum of diseases with reference to known past cases, as well as treatment modalities available for *Nocardia transvalensis* infections.

### 1.1. Taxonomy and Microbiological Description

*Nocardia* species are Gram-positive, catalase-positive, urease-positive, aerobic, partially acid-fast, filamentous, slow-growing, rod-shaped, and beaded, actinomycetes ubiquitously found in the environment world-wide, and they are predominantly observed in soil in warm climates [1,2,3]. There does appear to be some geographic variability in the prevalence of *Nocardia* species worldwide; however, the majority of *Nocardia* species are known to be universally prevalent [4]. *Nocardia* is partially acid-fast due to traditional carbol-fuchsin staining and due to the mycolic acid being present within the cell wall. *Nocardia’s* characteristic “beaded” pattern is often the distinguishing factor from other mycobacterial species that stain well on acid-fast staining [1,2,3].

*Nocardia* taxonomy continues to develop, with more than 50 species that have been clinically identified and that are responsible for disease pathogenesis. The U.S. Centers for Disease Control and Prevention (CDC) denotes *Nocardia nova*, *Nocardia farcinica*, *Nocardia cyriacigeorgica*, *Nocardia brasiliensis*, and *Nocardia abscessus* as the most frequently isolated organisms and clinically encountered species of *Nocardia* in the U.S. [5]. *Nocardia transvalensis* complexes, which include *Nocardia blacklockiae*, *Nocardia wallacei*, and, relevant to this case- report, *Nocardia transvalensis,* are more rarely described [2]. Deciphering between members of the *Nocardia transvalensis* complex is best attained through gene sequencing methods, as phenotypic biochemical depictions are time-consuming and not as reliable [4].

*Nocardia transvalensis* virulence is low, so disease burden is generally prominent in immunosuppressed patients or those with weakened systems. The virulence enzymes catalase and superoxide dismutase inactivate reactive oxygen species emitted by host neutrophils, which are toxic to bacteria. Chronic granulomatous diseases are thus more prone to *Nocardia* infections. Equally important to its virulence is the cord factor responsible for preventing the fusion of phagosomes with lysosomes [6,7].

### 1.2. Epidemiology

*Nocardia transvalensis* is a rare cause of human nocardiosis and accounts for a minority of *Nocardia* infections globally, with only a few documented case reports published. World-wide, the incidence of general *Nocardia* infections is not known but is thought to be increasing given the simultaneously increasing rate of immunocompromised patients [8]. Several studies have been completed to demonstrate local clinical distribution by species of *Nocardia*. One study from Spain published in the *Journal of Antimicrobial Chemotherapy* evaluated over 1100 strains of *Nocardia* clinically identified between 2005 and 2014, finding an incidence rate of approximately 2.4% [9]. Another study published in a 2024 BMC Microbiology journal analyzed the application of matrix-assisted laser desorption ionization time-of-flight mass spectrometry (MALDI-TOF MS) in identifying *Nocardia* species in China, isolating different strains of *Nocardia* between 2016 and 2022, with approximately only 1% of strains reported as *Nocardia transvalensis* [10].

Given the scarcity of the literature, *Nocardia transvalensis* is implied to infect similar patient populations as other *Nocardia* species. Nocardiosis typically manifests as an opportunistic infection in patients with compromised immune systems from pre-existing diseases such as cancer, HIV, chemotherapy, transplant recipients, or chronic corticosteroid therapy. Only in a minority of instances does *Nocardia* manifest in immunocompetent patients, and of the cases, only a cutaneous infection is present [1].

### 1.3. Transmission

*Nocardia transvalensis* is presumed to follow the same transmissibility patterns as other *Nocardia* species. The general route of inoculation primarily occurs through the inhalation of soil, dust, and other decaying organic matter, and it classically manifests as a pulmonary infection. Secondary localizations in the central nervous system and subcutaneous tissues can also occur. Less commonly, the direct inoculation of contaminated material into open wounds or traumatized skin can lead to cutaneous infectious, including mycetomas. However, this is not the predominant route of infection for *Nocardia transvalensis*. Person-to-person transmission has not been described in the medical literature; while immunocompromised patients are more susceptible to exposures, transmission remains primarily environmental rather than nosocomial or interpersonal [11].

### 1.4. Diagnosis

Like other *Nocardia* species, the diagnosis of *Nocardia transvalensis* is established primarily through culture from the infection site. Standard samples entail sputum and broncheoalveolar lavage, skin biopsies, abscess aspirates, and cerebrospinal fluid from pulmonary disease, cutaneous disease, abscesses, and central nervous system infections, respectively. Additionally, in cases of suspected dissemination, blood cultures are diagnostic for *Nocardia transvalensis* bacteremia [12].

For pulmonary disease, imaging via chest CT is imperative in evaluating pulmonary *Nocardia transvalensis* infections. A wide variety of findings can be detected on pulmonary imaging, including cavitary lesions, diffuse pulmonary infiltrates, lung abscesses, or pleural effusions. In all patients with the presence of *Nocardia transvalensis*, central nervous system imaging must be undertaken through a brain CT or MRI. Central nervous system infections are the most common metastatic inoculation site, occurring in more than 40% in some studies [13]. Central nervous system infections can present primarily as a brain abscess or, rarely, meningitis [14].

### 1.5. Disease Spectrum and Previous Rare Cases

*Nocardia* infection presents through a spectrum of disease, but it most commonly involves the lung, skin and soft tissues, and central nervous system, as implied earlier. Pulmonary nocardiosis classically presents with respiratory distress, fever, dyspnea, and occasionally hemoptysis, with pulmonary findings on CT imaging. Cutaneous nocardiosis presents with either subcutaneous nodules, abscesses, cellulitis, or mycetomas. Central nervous involvement presents symptoms of brain abscess or meningitis along with focal neurologic deficits and seizures. Lastly, disseminated nocardiosis can infect virtually any organ [13].

There are very few published case reports of *Nocardia transvalensis* infection.

In the early 1900s, the first case of *Nocardia transvalensis* was reported through the presentation of a mycetoma in South Africa. A case described in the *American Journal of Tropical Medicine and Hygiene* reported a case of mycetoma pedis extending over the malleoli in which cultures obtained on blood-agar pathogenically resembled *Nocardia* but differed phenotypically from other species of *Nocardia*. It was implied that this mycetoma was attributed to *Nocardia transvalensis* [15].

In 1995, a few more cases of *Nocardia transvalensis* were reported. A case report article published in the *European Journal of Clinical Microbiology and Infectious Diseases* illustrated one of these cases in which a patient had disseminated infection with pulmonary involvement of *Nocardia transvalensis* after a liver transplant. This patient responded well to trimethoprim-sulfamethoxazole therapy, but the clinical rarity of the microbe and the lack of known antibiotic sensitivities were highlighted [16].

In a 2023 case report published in the *New Microbes and New Infection Journal*, the first case of community-acquired pneumonia with asymptomatic disseminated brain abscess due to *Nocardia transvalensis* and *Nocardia farcinica* was reported. This was the first and only case reported thus far of *Nocardia transvalensis* infecting an immunocompetent patient that had successful treatment initially with triple empiric therapy with meropenem, trimethoprim-sulfamethoxazole, and amikacin, followed by targeted therapy with trimethoprim-sulfamethoxazole, moxifloxacin, and folinoral for 6 months. Brain imaging reverted to normal findings post-therapy; however, this patient was left with symptomatic and radiographic pulmonary sequelae [2].

The largest reported clinical database of *Nocardia transvalensis* comes from a 1992 case series article in the *Clinical Infectious Disease Journal*, illustrating a retrospective review from 1981 to 1990, identifying only 15 cases world-wide of *Nocardia transvalensis*. Even within these 15 cases, only 10 patients had clinical disease, while many of the other patients had colonization. Six of these eight patients had pulmonary or disseminated *Nocardia transvalensis* infections, had an underlying immunologic disorder, and were receiving immunosuppressive medications. Results of antimicrobial susceptibilities were only able to be performed on some isolates of *Nocardia transvalensis*, but found increasing resistance to amikacin compared to other more common strains of *Nocardia* [17].

### 1.6. Treatment Modalities

The mainstay of antimicrobial treatment for *Nocardia* infections has historically been sulfonamides, with the most common antibiotic being trimethoprim-sulfamethoxazole in the U.S. [18]. Resistance to sulfonamides is increasing amongst *Nocardia* species, so combination therapy with carbapenems, fluoroquinolones, oxazolidinones, or tetracyclines is often employed. If central nervous system involvement is present, ceftriaxone in conjunction with trimethoprim-sulfamethoxazole is utilized [19].

Sensitivity data for *Nocardia transvalensis* are scarce, with only a few reports illustrating antimicrobial sensitivities.

According to a 2014 *Antimicrobial Agents and Chemotherapy* article identifying antimicrobial susceptibility among clinical *Nocardia* species, only one strain of *Nocardia transvalensis* was isolated out of 149 *Nocardia* species (unspecified primary infection site). The available susceptibility data of this multi-drug-resistant organism demonstrated resistance to amikacin, amoxicillin-clavulanic acid, cefepime, clarithromycin, doxycycline, minocycline, and tobramycin. Sensitivities were only noted for trimethoprim-sulfamethoxazole, ceftriaxone, ciprofloxacin, linezolid, and moxifloxacin [20].

Likewise, a 2017 study in the *Journal of Antimicrobial Chemotherapy* evaluated *Nocardia* susceptibility patterns over a 10 year period in Spain. Over 1000 *Nocardia* isolates were analyzed, with only 2.4% of these species being *Nocardia transvalensis*. Susceptibilities to antibiotics were determined using the Etest. The majority of *Nocardia transvalensis* isolates were resistant to tobramycin, erythromycin, and minocycline. Interestingly, approximately 40–50% of all *Nocardia transvalensis* isolates were resistant to trimethoprim-sulfamethoxazole and amikacin [9].

## 2. Case

A 45-year-old homeless male with a past medical history of resolved hepatitis C with recent undetectable viral loads, intravenous (IV) drug use, alcohol use disorder, and tobacco use of 30 years presented to a level-one trauma center emergency department with complaints of lower back pain, cough, fatigue, and dyspnea. The patient noted these symptoms spontaneously developing shortly after exposure to rain while sleeping beside a lake. Of note, the patient had been previously hospitalized three weeks earlier in a different tertiary hospital and left that facility against medical advice.

While inpatient at his initial hospitalization, the patient was found to have right upper lobe pneumonia and was treated with IV cefepime 2 g every 8 h and IV vancomycin as dosed by trough levels. Initial infectious workup with blood cultures, respiratory cultures, and viral respiratory panel was unremarkable. An MRI with contrast of the lumbar spine was obtained to evaluate his acute back pain, which demonstrated soft tissue edema and myositis with a possible abscess and subtle osteomyelitis of the L3–L4 facet joint and transverse process. Given these findings, the patient underwent a CT-guided aspiration of the paraspinal abscess by interventional radiology. Aspirate cultures were identified via MALDI-TOF MS that resulted in *Nocardia transvalensis* being identified as the causative organism. The patient was transitioned to IV trimethoprim-sulfamethoxazole 400 mg every 8 h, IV ceftriaxone 2 g every 12 h, and IV linezolid 600 mg every 12 h but left against medical advice shortly afterward.

Upon re-admission for persistent symptoms in his second hospitalization, the patient’s vitals were: blood pressure of 170/92 mmHg, heart rate of 79 beats/min, and temperature of 36.7 °C. Lab workup was notable for lactic acid of 1.7 mmol/L (normal: 0.4–2.0 mmol/L), white blood cell count of 6.0 × 10^9^/L (normal: 4.0–11 × 10^9^/L), and procalcitonin of 0.10 ng/mL (normal: <0.05 ng/mL); a CT head with and without contrast revealed 9 mm enhancing supratentorial lesions along the splenium of the corpus callosum (Figure 1). A repeat MRI of the lumbar spine revealed concerns for a neoplastic process from the spinous process of L2 with additional involvement of the bilateral inferior articular processes, left L3 pedicle, and left transverse process (Figure 2). The reviewing radiologist favored lymphoma with additional involvement of the bilateral inferior articular process. Neurosurgery consultation deemed no significant interventions for his cranial lesions necessary. However, the patient underwent an IR-guided biopsy of the spinal lesion, ultimately yielding only granulation tissue without evidence of malignancy. Given his housing instability, the patient’s final antibiotic regimen based on susceptibility pattern identified via broth microdilution and feasibility entailed oral levofloxacin 750 mg twice daily, oral linezolid 600 mg twice daily, and oral trimethoprim-sulfamethoxazole 800 mg–160 mg three times daily for a course of 60 days with planned outpatient follow-up in the infectious disease clinic (Table 1). Unfortunately, the patient refused follow-up as an outpatient after discharge and terminated all means of communication.

## 3. Discussion

The patient in this case initially presented with pulmonary symptoms and generalized constitutional symptoms, with later detection of central nervous system involvement after further diagnostic interrogation to assess an unclear etiology of his back pain. The unifying diagnosis to explain the compilation of symptoms he had is *Nocardia transvalensis* infection. There are several essential discussion points to illustrate in this case.

The disease progression in this patient likely followed that which is typical of *Nocardia* infections. This patient with pulmonary-spinal symptoms likely contracted *Nocardia transvalensis* through inhalation of the organism from outdoors. His homelessness status, living next to a fresh-water lake, and being vulnerable to heavy rain likely exposed him to a warm and humid climate, makings him susceptible to *Nocardia* infection [11]. His disease likely started with respiratory symptoms alone, characterized by a lobar pneumonia, followed by dissemination to involve his central nervous system, presenting as lumbar back pain with a paraspinal abscess. There was no significant evidence of other disseminated disease in this case. Had standard precautions with sufficient housing been available to this patient, he likely would not have contracted this infection.

The differential diagnosis considerations and clinical workup were also sufficient to ultimately arrive at the diagnosis of this rare disease. Within the workup of this patient with pulmonary-spinal symptoms attributed to *Nocardia transvalensis*, a multitude of differentials were entertained and investigated appropriately. Lung malignancy was heavily considered amongst all medical teams treating the patient, given the pulmonary imaging, his tobacco abuse history, and concern for metastasis to his central nervous system. Mycobacterial infections, including tuberculosis, were also of concern given the location of the pneumonia, his homelessness and alcohol use disorder, and initial positive findings on acid-fast staining. Standard pulmonary pneumonias were considered, given the respiratory symptoms and frequency of this diagnosis in the hospital. Lastly, other fungal infections, including aspergillosis, histoplasmosis, and blastomycosis, were discussed, given historic exposures to endemic regions and pulmonary-spinal findings. All these differentials, however, were eliminated as potential diseases after biopsy results demonstrated *Nocardia transvalensis.*

The management in this case likewise matched the pace at which diagnostic information was available to us. Initially, broad-spectrum antibiotics with vancomycin and cefepime were appropriately given for presumed pneumonia. While his pulmonary workup was relatively unfruitful, what clued us in to the diagnosis was the patient’s history of complaining of incidental back pain. While musculoskeletal pain is a common complaint in inpatient and outpatient settings, this patient attained dedicated and proper attention to his subacute lumbar back pain through formal imaging. It was by detecting subtle osteomyelitis of the L3-L4 facet joint and transverse process with a concerning paraspinal abscess that a biopsy was able to be performed. When the patient left his first hospitalization against medical advice, antibiotic susceptibilities were not available at the time. The empiric regimen was transitioned to trimethoprim-sulfamethoxazole, ceftriaxone, and linezolid in response to the new detection of *Nocardia transvalensis*, which was very appropriate given the susceptibility data in the limited literature, need for a regimen with high penetrance to central nervous system tissue, and the final susceptibilities of this strain based on broth microdilution [9,19,20].

This case report is very impactful for numerous reasons.

First, this case adds to the minimal global repertoire of the literature on pathogenic *Nocardia transvalensis*. As mentioned before, there are only a few cases of *Nocardia transvalensis* that have been reported, with the most extensive study reporting clinical manifestation in roughly 10 cases world-wide [17]. Our patient behaved similarly to the majority of these reported cases by presenting with both pulmonary-neurologic clinical manifestations [17]. Furthermore, only a minority of these cases resulted in fatal outcomes; our patient followed the majority of cases, with no fatal consequences [17].

Second, and what is most unique about this case, is *Nocardia transvalensis* manifesting in an immunocompetent host. Of the limited cases available, *Nocardia transvalensis* has been known to be a disease in immunocompromised hosts [11]. The patient in this case, however, had no identifiable immunocompromised condition that predisposed him to *Nocardia transvalensis* disseminated infections. Appropriate investigations through lumbar biopsies and serological testing were all pursued to uncover any potential etiologies of immunocompromise or malignancy, but they all ultimately proved clinically unremarkable. From our research, this appears to be only the second known case of *Nocardia transvalensis* infecting an immunocompetent host [2].

Third, our case provides additional antimicrobial susceptibility data to a limited reported data set, given the rarity of *Nocardia transvalensis*. Identifying the species of *Nocardia* is critical due to the differing spectrum of susceptibility patterns. Gene sequencing with the 16S rRNA gene or MALDI-TOF MS is primarily used to identify *Nocardia* species [21]. Our patient’s *Nocardia* species was identified primarily through MALDI-TOF MS. Sensitivities were primarily obtained utilizing broth microdilution and were noted to be only susceptible to trimethoprim-sulfamethoxazole, fluoroquinolones, linezolid, imipenem, and ceftriaxone, which not only limited our antimicrobial options conducive to the patient’s social dilemma in the setting of housing instability, difficulties with medication adherence, and intravenous drug use but also differed from the typically reported susceptibility patterns found with *Nocardia transvalensis*.

## 4. Conclusions

This case overall exemplifies a rarely encountered clinical scenario of *Nocardia transvalensis.* Having discussed the current nature of the available literature contextualizing *Nocardia transvalensis*, the details surrounding our unique patient’s case, and clinical significance of the case within our discussion, there are several final conclusive points for clinicians to ultimately take from this report.

First, this case illustrates how one should clinically recognize active *Nocardia transvalensis* infection. Given the rarity of *Nocardia transvalensis*, there are no clear differing pathogenic mechanisms noted between strains of *Nocardia transvalensis* or other species of *Nocardia*. The disease pattern of *Nocardia transvalensis* typically follows that of other Nocardia organisms [13]. Patients presenting with cutaneous disease with either subcutaneous nodules, abscesses, cellulitis, or mycetomas and pulmonary symptoms with symptoms and radiological imaging of pneumonia should consider *Nocardia* differentials in their workup, especially if they are immunocompromised and/or have environmental risk factors. Furthermore, when *Nocardia* is suspected or confirmed, interrogative diagnostics must be undertaken to rule out central nervous system dissemination [13].

Second, the diagnosis of *Nocardia transvalensis* encompasses a combination of clinical, radiological, and microbiological assets. Clinical suspicion, as described above, can lead to dedicated imaging of the affected organs. Evidence of pneumonia, lung abscess, pleural fluid accumulation, brain abscess, or spinal cord abscess can be targeted for aspiration and sent off for species identification via MALDI-TOF MS, as in our patient, or gene sequencing with 16S rRNA [21]. Antibiotic susceptibility patterns can further be generated via broth microdilution, as practiced in our case. Antibiotic susceptibility patterns documented for *Nocardia transvalensis* are rare, but our case illustrates a strain that was most sensitive to trimethoprim-sulfamethoxazole, clarithromycin, fluoroquinolones, linezolid, imipenem, and ceftriaxone.

Lastly, this case very clearly demonstrates how *Nocardia transvalensis* can still present in patients with no identifiable immunocompromised conditions. While *Nocardia transvalensis* is highly correlated with immunocompromised conditions, it is not absolute.

## Figures and Tables

**Figure 1 idr-17-00140-f001:**
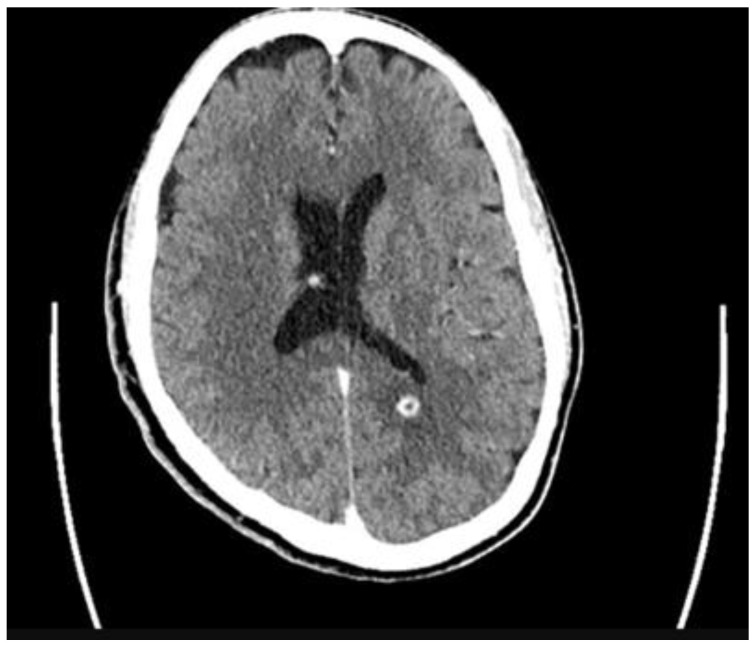
Supratentorial lesions on CT head.

**Figure 2 idr-17-00140-f002:**
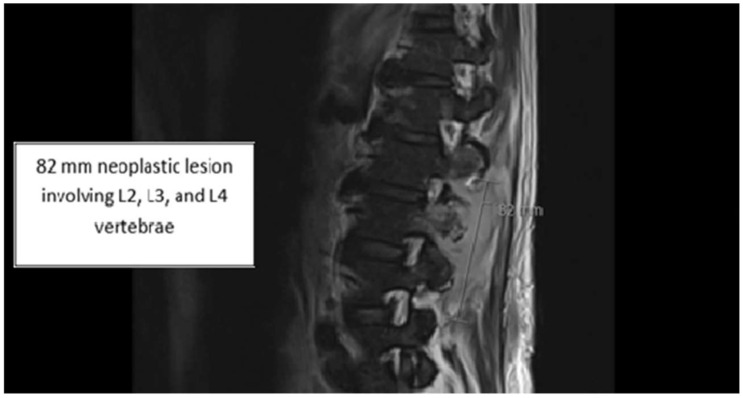
Concerns of neoplastic processes on spinal MRI.

**Table 1 idr-17-00140-t001:** Susceptibility patterns of *Nocardia Transvalensis*.

**Gram Stain:** Few gram positive branching bacilli	
**Results:** Many Nocardia sp. transvalensis	
**Method:** MIC reference susceptibility	
**Antibiotic**	**Susceptibility (µg/mL)**
Amikacin	16: Resistant
Amoxicillin/clavulanate	32: Resistant
Ceftriaxone	4: Susceptible
Ciprofloxacin	0.5: Susceptible
Clarithromycin	0.5: Susceptible
Doxycycline	4: Intermediate
Imipenem	0.5: Susceptible
Linezolid	1: Susceptible
Minocycline	2: Intermediate
Moxifloxacin	0.12: Susceptible
Tobramycin	32: Resistant
Trimethoprim/Sulfamethoxazole	1: Susceptible

## Data Availability

Data analyzed during the study is available from the authors upon request.

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
