# Peer review of "A Rare Case of Disseminated Nocardia transvalensis in an Immunocompetent Host"

_2036-7449, 2025, doi:10.3390/idr17060140_

Round 1
Reviewer 1 Report
Comments and Suggestions for Authors
In the Discussion section, particularly on line 230, the term "spores" is used; nocardia do not exhibit this type of structure.
It is extremely important to include more detailed information about how you identified Nocardia transvaliensis. Did you use biochemical tests or automated systems? Include this information on line 196.
It would be best if the Informed Consent Statement was provided in a physical form with a handwritten signature, in addition to being recorded in their electronic medical system.
Author Response
Comment 1: In the Discussion section, particularly on line 230, the term "spores" is used; nocardia do not exhibit this type of structure.
Response 1: Agreed. We have eliminated the word spores
Comment 2: It is extremely important to include more detailed information about how you identified Nocardia transvaliensis. Did you use biochemical tests or automated systems? Include this information on line 196.
Response 2: Agreed. We have added both in the case as well as in the discussion and conclusion now that MALDITOF MS and broth microdilution were the microbiological tools used to diagnose this species.
Comment 3: It would be best if the Informed Consent Statement was provided in a physical form with a handwritten signature, in addition to being recorded in their electronic medical system.
Response 3: The patient provided verbal consent and was aware of documentation of the verbal consent and intent to submit for publication. He was not able to sign a consent form for MDPI.

Reviewer 2 Report
Comments and Suggestions for Authors
This study reports a rare case of infection caused by Nocardia transvalensis in humans. This is very important for human health. This report provides a detailed account of the case situation, but contains very little information about the biological characteristics of the bacteria causing the infection.
Suggestions:
- The authors mentioned that Nocardia transvalensis was identified in this case. But the reports didn’t show this data, and there should be an analysis of the biological evolution tree and the sequencing of the bacterial genome.
- Is there any difference in the pathogenicity of the strain this patient is infected with compared to other species?
- So far, only a few cases of nocardia infection have been reported. Globally, approximately 10 patients have exhibited clinical symptoms. What are the similarities and differences between this case and the cases that have been reported?
- What is the most likely cause of the patient's infection with Nocardia transvalensis? What are the precautions that need to be taken for normal individuals?
Author Response
Comment 1: The authors mentioned that Nocardia transvalensis was identified in this case. But the reports didn’t show this data, and there should be an analysis of the biological evolution tree and the sequencing of the bacterial genome.
Response 1: Our case report primarily focuses on clinical aspect of Nocardia. We do not have evidence of the sequencing of the bacterial genome.
Comment 2: Is there any difference in the pathogenicity of the strain this patient is infected with compared to other species?
Response 2: No difference noted in pathogenicity. We created a conclusion section to depict this.
Comment 3: So far, only a few cases of nocardia infection have been reported. Globally, approximately 10 patients have exhibited clinical symptoms. What are the similarities and differences between this case and the cases that have been reported?
Response 3: After this comment, we have added in more to our first point of our discussion section. We compared the patients of the 10 patients to this case.
Comment 4: What is the most likely cause of the patient's infection with Nocardia transvalensis? What are the precautions that need to be taken for normal individuals?
Response 4: We do outline the likely mechanism by which the patient got the infection in the discussion section. We enhance this by stating that had he had shelter, he likely would not have gotten the infection, which alludes to the standard precautions most people need to take.

Reviewer 3 Report
Comments and Suggestions for Authors
Thank you for inviting me to review this manuscript. It is interesting. I have some comments that might be of use:
- Line 15 and 21: All names of microorganisms should be in italics. Please correct
- Line 17 and 19: All abbreviations should be written in full when first mentioned (MRI and CT in this occasion)
- The abstract section is very small. Please expand by giving a couple of lines as an introduction and more information about treatment
- Lines 26-32 lack references
- I would suggest removing the subheadings in the introduction section
- Three and a half pages are relatively long for an introduction, even though the information stated is important. I suppose it could be somewhat reduced
- Figure 1: please improve image quality. The words at the left side of the image should be removed. Also, more information should be given, such as ‘computed tomography of the head with intravenous contrast showing…’
- The legend of Figure 2 also needs improvement. Is that T1? T2? It is not clear. It should be more descriptive and informative
- Table 1: the numbers at the right column should have a measurement value. Is that μg/ml? Also, I wonder if the second column is needed
- For how long was the patient treated? Any data about follow up?
- A conclusion subsection should be added at the end of the discussion section summarizing the key points of this case report
- Details on microbiology should be mentioned for the case presented, such as way of identification (MALDI-TOF? Sequencing?)
- Please mention the doses for the antimicrobials that were used
Author Response
Comment 1: Line 15 and 21: All names of microorganisms should be in italics. Please correct
Response 1: We have corrected the italics
Comment 2: Line 17 and 19: All abbreviations should be written in full when first mentioned (MRI and CT in this occasion)
Response 2: We have corrected the abbreviations by labeling them initially as full titles.
Comment 3: The abstract section is very small. Please expand by giving a couple of lines as an introduction and more information about treatment
Response 3: More of the abstract was expanded by adding a few lines on an introduction and treatment of the patient.
Comment 4: Lines 26-32 lack references
Response 4: Addressed now.
Comment 5: I would suggest removing the subheadings in the introduction section
Response 5: We do disagree slightly here. The subheadings make for a good logical flow of information and easy to find sections. We would prefer keeping the subheadings in place.
Comment 6: Three and a half pages are relatively long for an introduction, even though the information stated is important. I suppose it could be somewhat reduced
Response 6: Noted. We were initially told that our introduction is too small. We have expanded it and have been as comprehensive as possible to outline all available literature out there and the impacts this case makes to the current literature now.
Comment 7: Figure 1: please improve image quality. The words at the left side of the image should be removed. Also, more information should be given, such as ‘computed tomography of the head with intravenous contrast showing…’
Response 7: The image has been improved by getting rid of the words. Within the body of the text, we have described the image more.
Comment 8: The legend of Figure 2 also needs improvement. Is that T1? T2? It is not clear. It should be more descriptive and informative.
Response 8: I have included a very small description within the image outlining what you are looking at. Moreover, a descriptive text of the image is included in the body of the paragraph above the image. Thanks for pointing this out.
Comment 9:Table 1: the numbers at the right column should have a measurement value. Is that μg/ml? Also, I wonder if the second column is needed.
Response 9: I redid the table. I got rid of the redundancy and placed in labels appropriately
Comment 10: For how long was the patient treated? Any data about follow up?
Response 10: I added a section in the case outlining the duration of treatment and follow up the patient had (or lack there of).
Comment 11: A conclusion subsection should be added at the end of the discussion section summarizing the key points of this case report.
Response 11: We have added a conclusion now.
Comment 12: Details on microbiology should be mentioned for the case presented, such as way of identification (MALDI-TOF? Sequencing?)
Response 12: We have added the details of the microbiological tactics used to diagnose the infection in the case and discussion now. It should be added.
Comment 13: Please mention the doses for the antimicrobials that were used
Response 13: Completed.

Round 2
Reviewer 2 Report
Comments and Suggestions for Authors
N/A
Reviewer 3 Report
Comments and Suggestions for Authors
The manuscript was improved